# 3D oxygen vacancy distribution and defect-property relations in an oxide heterostructure

Kasper A. Hunnestad [1,9,10], Hena Das[2], Constantinos Hatzoglou[1], Megan Holtz [3,4], Charles M. Brooks[4], Antonius T. J. van Helvoort [5], David A. Muller [3,6], Darrell G. Schlom [4,6,7], Julia A. Mundy [8] & Dennis Meier [1] ✉

Oxide heterostructures exhibit a vast variety of unique physical properties. Examples are unconventional superconductivity in layered nickelates and topological polar order in $(PbTiO_3)_n/(SrTiO_3)_n$ superlattices. Although it is clear that variations in oxygen content are crucial for the electronic correlation phenomena in oxides, it remains a major challenge to quantify their impact. Here, we measure the chemical composition in multiferroic $(LuFeO_3)_9/(LuFe_2O_4)_1$ superlattices, mapping correlations between the distribution of oxygen vacancies and the electric and magnetic properties. Using atom probe tomography, we observe oxygen vacancies arranging in a layered three-dimensional structure with a local density on the order of $10^{14}\,cm^{-2}$, congruent with the formula-unit-thick ferrimagnetic $LuFe_2O_4$ layers. The vacancy order is promoted by the locally reduced formation energy and plays a key role in stabilizing the ferroelectric domains and ferrimagnetism in the $LuFeO_3$ and $LuFe_2O_4$ layers, respectively. The results demonstrate pronounced interactions between oxygen vacancies and the multiferroic order in this system and establish an approach for quantifying the oxygen defects with atomic-scale precision in 3D, giving new opportunities for deterministic defect-enabled property control in oxide heterostructures.

The concentration and distribution of oxygen in strongly correlated electron systems is essential for the material's response[1,2]. By introducing oxygen vacancies or interstitials, electronic and magnetic properties can be controlled, and even entirely new functional properties can be obtained[3]. For example, redox reactions can change the oxygen-stoichiometry and drive topotactic transitions[4], resistive switching[5], and ferroelectric self-poling[6]. In structured materials, the oxygen diffusion length is usually comparable to the dimensions of the system[7] and local variations in oxygen content naturally arise due to varying defect formation energies[8]. The latter plays a crucial role for

[1]Department of Materials Science and Engineering, NTNU Norwegian University of Science and Technology, Trondheim 7491, Norway. [2]Laboratory for Materials and Structures, Institute of Innovative Research, Tokyo Institute of Technology, 4259 Nagatsuta, Midori-ku, Yokohama 226-8503, Japan. [3]School of Applied and Engineering Physics, Cornell University, Ithaca, NY 14853, USA. [4]Department of Materials Science and Engineering, Cornell University, Ithaca, NY 14853, USA. [5]Department of Physics, NTNU Norwegian University of Science and Technology, Trondheim 7491, Norway. [6]Kavli Institute at Cornell for Nanoscience, Ithaca, NY 14853, USA. [7]Leibniz-Institut für Kristallzüchtung, Max-Born-Str. 2, Berlin 12489, Germany. [8]Department of Physics, Harvard University, Cambridge, MA 02138, USA. [9]Present address: Acoustics Group, Department of Electronic Systems, NTNU Norwegian University of Science and Technology, Trondheim 7491, Norway. [10]Present address: Centre for Geophysical Forecasting, NTNU Norwegian University of Science and Technology, Trondheim 7491, Norway. ✉e-mail: dennis.meier@ntnu.no

property-engineering in oxide heterostructures, where atomically precise interfaces in combination with defect engineering are used to tailor, e.g., polar order[9–11], magnetic exchange interactions[12], and the onset of superconductivity[13–15].

Quantifying emergent spatial variations in oxygen content at the atomic level, however, is extremely challenging[1,16]. Enabled by the remarkable progress in high-resolution transmission electron microscopy, it is possible to image individual oxygen defects in heterostructures[17] and, for sufficiently high defect densities, chemical fingerprints associated with their accumulation or depletion at interfaces/interlayers can be detected[18–21]. Despite their outstanding capabilities, these electron-microscopy based methods are not quantitative and inherently restricted to 2D projections along specific zone axes. This restriction prevents the full three-dimensional (3D) analysis of oxygen defects and limits the microscopic understanding of the interplay between oxygen defects and the material's physical properties. An experimental approach that, in principle, facilitates the required chemical accuracy and sensitivity to overcome this fundamental challenge is atom probe tomography (APT). The potential of APT is demonstrated by previous work on bulk oxide superconductors[2] and ferroelectrics[22], measuring stoichiometric variations at the nanoscale and lattice positions occupied by individual dopant atoms, respectively.

Here, we quantify the distribution of oxygen vacancies in $(LuFeO_3)_9/(LuFe_2O_4)_1$ superlattices and demonstrate a correlation with the electric and magnetic orders that lead to room-temperature multiferroicity in this system. Using APT, we show that oxygen vacancies ($V_O$) have a propensity to accumulate in the $LuFe_2O_4$ monolayers, forming a layered 3D structure with an average density of about $(7.8 \pm 1.8) \times 10^{13}$ cm$^{-2}$. The oxygen vacancies facilitate the electrical screening that is essential for stabilizing the ferroelectric order and control the oxidation state of the iron (Fe), which is responsible for the emergent ferrimagnetism. The results clarify the defect-property relation and show that the multiferroic behavior in $(LuFeO_3)_9/(LuFe_2O_4)_1$ is intertwined with, and promoted by, the 3D oxygen vacancy order.

## Results and discussion

Figure 1a shows a high-angle annular dark-field scanning transmission electron microscopy (HAADF-STEM) image of the $(LuFeO_3)_9/(LuFe_2O_4)_1$ superlattice. The system exhibits spontaneous electric and magnetic order, facilitating magnetoelectric multiferroicity at room temperature[23]. The ferroelectricity relates to the displacement of the Lu atoms in the $LuFeO_3$ layers (up-up-down: +$P$; down-down-up: -$P$, see Fig. 1a), whereas the ferrimagnetism has been explained based on $Fe^{2+} \rightarrow Fe^{3+}$ charge-transfer excitations in the $LuFe_2O_4$ layers[24]. Interestingly, the multiferroic $(LuFeO_3)_9/(LuFe_2O_4)_1$ superlattice develops an unusual ferroelectric domain state with extended positively charged domain walls in the $LuFeO_3$ layers, where the polarization meets head-to-head ($\rightarrow\leftarrow$)[25]. The formation of charged head-to-head domain walls is surprising as they have high electrostatic costs, which raises the question how the material stabilizes them.

To understand the microscopic mechanism that leads to the distinct magnetic and electric order in $(LuFeO_3)_9/(LuFe_2O_4)_1$, we map the 3D chemical composition of the superlattice using APT. For the APT analysis, we deposit a protective capping layer (Pt, Cr or Ti) and prepare needle-shaped specimens using a focused ion beam (FIB, see Methods) as shown in Fig. 1b. The needle-like shape is a requirement in APT experiments and allows for producing the high electric fields required for field evaporation of surface atoms when a voltage > 2 kV is applied. The capping layer ensures that the $(LuFeO_3)_9/(LuFe_2O_4)_1$ superlattice is located below the apex of the needle, which enables us to analyze a larger volume and, hence, improve the chemical precision of the experiment. Figure 1c shows the 3D reconstruction of the measured volume, where the Fe and ZrO ionic species are presented

to visualize the superlattice and substrate, respectively (mass spectrum and bulk chemical composition are presented in Fig. S1). The $LuFe_2O_4$ layers are visible as darker lines due to their higher density of Fe atoms compared to $LuFeO_3$. The 3D reconstruction shows that the spacing between the $LuFe_2O_4$ layers varies within the analyzed volume of the superlattice, ranging from ≈2 nm to ≈6 nm. At the atomic scale, the $LuFe_2O_4$ layers exhibit the characteristic double-Fe columns (Fig. 1d), consistent with the HAADF-STEM data in Fig. 1a. Furthermore, enabled by the 3D APT imaging, we observe step-like discontinuities in individual $LuFe_2O_4$ layers in Fig. 1c. The observation of such growth-related imperfections leads us to the conclusion that the multiferroic response of the material is rather robust and resilient against such local disorder.

Most importantly for this work, the APT measurement provides information about the local chemical composition of the superlattice. Figure 2a displays the concentration of the different atomic species evaluated for the region marked by the white dashed line in Fig. 1c. The line plots are derived by integrating the data in the direction perpendicular to the long axis of the needle-shaped sample, showing pronounced anomalies at the position of the $LuFe_2O_4$ layers (marked by dashed lines). In total, seven peaks are resolved labeled ① to ⑦; two peaks correspond to the discontinuous $LuFe_2O_4$ layer (represented by the double-peak ①/②), and five peaks to the continuous $LuFe_2O_4$ layers resolved in Fig. 1c (③ to ⑦). In all cases, we consistently find an enhancement in Fe concentration and a decrease in Lu and O concentration in the $LuFe_2O_4$ layers relative to $LuFeO_3$. For each layer, more than $10^5$ atoms are measured, over a region of $20 \times 20$ nm$^2$, providing local error estimates for the concentrations on the order of

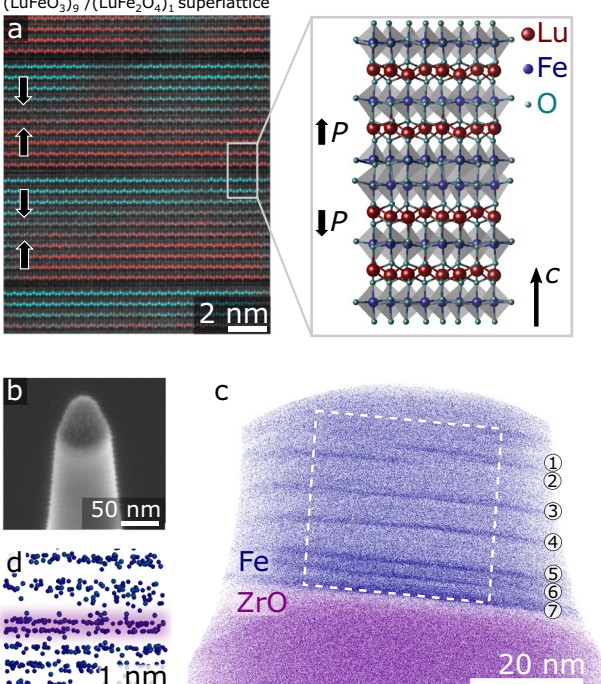

**Fig. 1 | 3D imaging of the $(LuFeO_3)_9/(LuFe_2O_4)_1$ superlattice structure. a** HAADF-STEM image recorded along the [100] zone axis and schematic, showing the atomic structure of the superlattice. Ferroelectric +$P$ and -$P$ domains are colored red and blue, respectively. **b** SEM image of an APT needle. Three different layers are visible, corresponding to the Cr protection layer (dark grey), the $(LuFeO_3)_9/(LuFe_2O_4)_1$ superlattice (bright), and the substrate. **c** 3D reconstruction of the APT data. Superlattice and substrate are represented by the Fe and ZrO ionic species, respectively. The dark lines in the superlattice correspond to double-Fe columns of the $LuFe_2O_4$ layers. **d** Zoom-in to one of $LuFe_2O_4$ layers in (**c**) resolving the double-Fe columns.

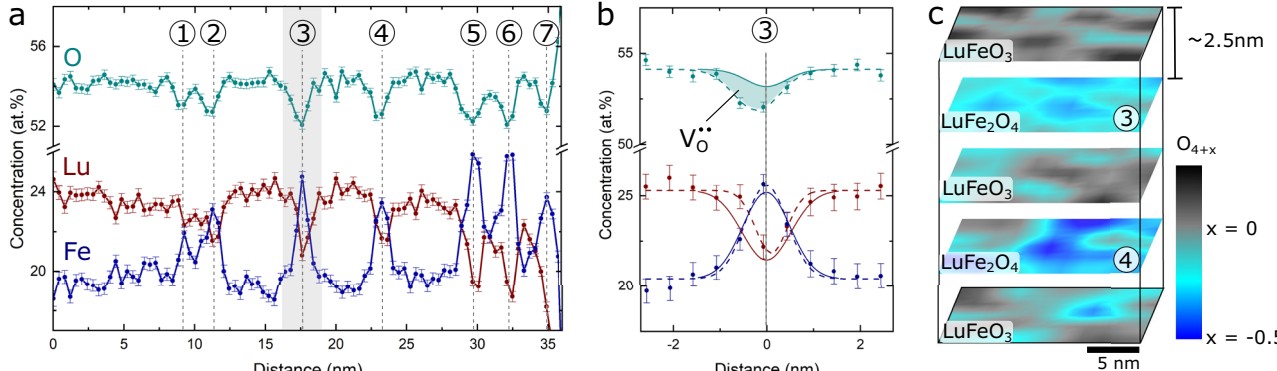

**Fig. 2 | 3D oxygen vacancy order. a** Profiles of the relative chemical composition, with the surface to the left in the plot. Error bars represent the standard deviation. Anomalies are observed at all the $LuFe_2O_4$ layers, numbered ① to ⑦. **b** Measured (data points) and theoretically expected (solid line) chemical concentration profile at $LuFe_2O_4$ layer ③. The shaded area highlights that the measured oxygen content is lower than in stoichiometric $LuFe_2O_4$, indicating an accumulation of oxygen vacancies. **c** 3D visualization of the oxygen stoichiometry based on the APT data set. Oxygen vacancies arrange in a layered three-dimensional structure congruent with the formula-unit-thick ferrimagnetic $LuFe_2O_4$ layers. Within the $LuFe_2O_4$ layers, oxygen vacancies form puddle-like regions of reduced $LuFe_2O_{4+x}$ (blue).

0.3 at.%. A more detailed analysis of the chemical composition of one of the continuous $LuFe_2O_4$ layers (i.e., layer ③) is presented in Fig. 2b. Figure 2b compares measured and calculated concentration profiles for Lu, Fe, and O. The calculated concentration profile corresponds to a stoichiometric $(LuFeO_3)_9/(LuFe_2O_4)_1$ superlattice, using a realistic experimental resolution of about 0.6 nm, showing a good agreement with the experimental data for Lu and Fe[26]. In contrast, the measured concentration of O is lower than expected, indicating an accumulation of oxygen vacancies, $V_O$. A small shift in the O profile relative to Fe and Lu is observed, which we attributed to field evaporation effects[27] (see also Fig. S3). By integrating over the layer, considering the difference between the measured and calculated profiles, we find a $V_O$ density of $(7.8 \pm 1.8) \times 10^{13}$ cm$^{-2}$, which corresponds on average to an oxygen-deficient state in $LuFe_2O_{4+x}$ with $x \approx -0.5$.

The same trend is observed for other $LuFe_2O_4$ layers with minor layer-to-layer variations in the $V_O$ density (see Fig. S2), indicating that the oxygen vacancies form a layered 3D structure within the $(LuFeO_3)_9/(LuFe_2O_4)_1$ superlattice, congruent with the arrangement of the $LuFe_2O_4$ layers. It is important to note, however, that within the different layers the distribution of $V_O$ is inhomogeneous as shown by the 3D map in Fig. 2c. This map presents the local chemical composition and reflects the average periodic variation in $V_O$ density in the $LuFeO_3$ and $LuFe_2O_4$ layers, consistent with the integrated data in Fig. 2a, b. Furthermore, it reveals a puddle-like distribution of the oxygen vacancies with puddle sizes in the order of a few nanometers and a maximum local $V_O$ density of up to $\approx 10^{14}$ cm$^{-2}$ (i.e., a reduction to $LuFe_2O_{3.5}$). In principle, our approach also allows for quantifying emergent layer-to-layer variations. The latter, however, requires a larger data set to gain statistically relevant information and is beyond the scope of this work.

To better understand the propensity of the oxygen vacancies to accumulate at the $LuFe_2O_4$ layers, we calculate and compare the $V_O$ defect formation energies for the $LuFeO_3$ and $LuFe_2O_4$ layers in Fig. 3a using density functional theory (DFT) calculations (Methods). Possible vacancy sites are located in the Lu- or Fe-layers ($V_O^{LuO_2}$ or $V_O^{FeO}$). The DFT calculations show that the formation of charged oxygen vacancies ($V_O^{\cdot\cdot}$) is energetically favorable compared to the neutral oxygen vacancies ($V_O^x$) leading to an energy reduction of about 1 eV (Fig. 3b). The lowest energy is achieved by incorporating the charged oxygen vacancies into the Fe-planes in $LuFe_2O_4$, reducing the energy by 64 meV compared to $LuFeO_3$. Figure 3c presented the respective defect formation energies as function of temperature. In summary, by accumulating charged oxygen vacancies in the $LuFe_2O_4$ layers, the $(LuFeO_3)_9/(LuFe_2O_4)_1$ superlattice can substantially reduce its energy, which promotes the

formation of $V_O^{FeO}$-rich $LuFe_2O_4$ layers and, hence, a layered 3D oxygen vacancy order consistent with the APT results.

The observed 3D oxygen vacancy order has a direct impact on the electric and magnetic properties and provides insight into their microscopic origin. As shown by the DFT calculations, the accumulation of $V_O$ effectively leads to electron-doping of the $LuFe_2O_4$ layers. We find that the locally measured $V_O$ density (Fig. 2) is equivalent to a positive charge of $25 \pm 5$ μC/cm$^2$. The latter explains why the $(LuFeO_3)_9/(LuFe_2O_4)_1$ superlattice can stabilize the unusual ferroelectric tail-to-tail configuration at the $LuFe_2O_4$ layers (seen in Figs. 1a and 3d). The polarization charges from the $LuFeO_3$ layers give a negative charge of about 13 μC/cm$^2$, which partially compensates the positive charge associated with the oxygen vacancies[28]. As a consequence of the energetically favored tail-to-tail configuration at the $LuFe_2O_4$ layers, formation of positively charged head-to-head domain walls within the $LuFeO_3$ layers is enforced which, in turn, are screened by a redistribution of mobile electrons generated by the $V_O$.

In conclusion, the ferroelectric properties of $(LuFeO_3)_9/(LuFe_2O_4)_1$ are found to be closely related to oxygen vacancy ordering. The oxygen vacancy order facilitates a transfer of electrons to the head-to-head domain walls in $LuFeO_3$ and drives the change in the oxidation state of Fe in the $LuFe_2O_4$ layers ($Fe^{2+} \rightarrow Fe^{3+}$), which is crucial for the ferrimagnetic order of the material[23–25]. The results clarify the microscopic origin of the unusual ferroelectric domain structure and provide an explanation for the enhancement of the magnetic response, revealing the importance of extrinsic defect-driven mechanisms for the emergence of room-temperature multiferroicity. Quantitative 3D imaging of oxygen defects and chemical profiling at the atomic scale is of interest beyond the defect-property relations discussed in this work and can provide insight into defect-driven effects and the general role that oxygen vacancies (or interstitials) play in emergent phenomena in oxide heterostructures. The approach shown demonstrates the benefit of quantitative atomic-scale characterization of oxygen at interfaces in oxides, which is crucial for a better understanding of their complex chemistry and physics, as well as improved property engineering and their utilization in nanoelectronic and oxitronic technologies.

## Methods
### Sample preparation and characterization
Thin films of $(LuFeO_3)_9/(LuFe_2O_4)_1$ were grown by reactive-oxide molecular-beam epitaxy in a Veeco GEN10 system on (111) $(ZrO_2)_{0.905}(Y_2O_3)_{0.095}$ (or 9.5 mol% yttria-stabilized zirconia) (YSZ)

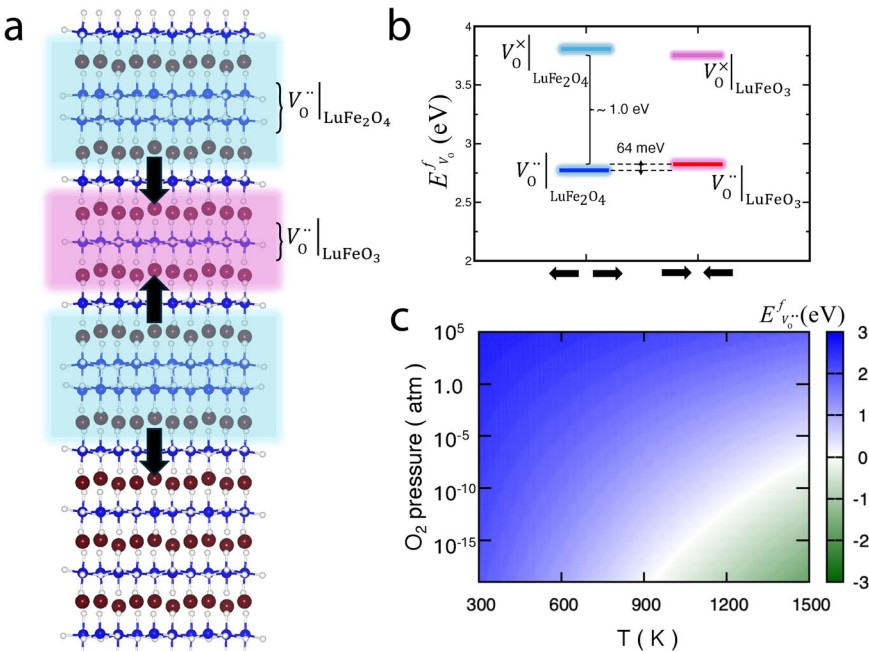

**Fig. 3 | Defect formation energy for oxygen vacancies in LuFe₂O₄ and LuFeO₃.** **a** Crystal structure of the $(LuFeO_3)_3/(LuFe_2O_4)_1$ superlattice, illustrating the local tail-to-tail and head-to-head polarization configurations (see also Fig. 1a). **b** The formation energy of oxygen vacancy at the oxygen-rich limit and for $E_F = 0$ (which corresponds to the valence band maximum). $V_O^{\cdot\cdot}|_{LuFe_2O_4}$ ($V_O^{\cdot\cdot}|_{LuFeO_3}$) and $V_O^{\times}|_{LuFe_2O_4}$ ($V_O^{\times}|_{LuFeO_3}$) correspond to oxygen vacancies in 2+ and neutral charge states in the LuFe₂O₄ (LuFeO₃) layer, respectively. Calculations are conducted for $U_{eff} = 6.5$ eV at the Fe 3d orbital and the corresponding band gap is $E_g = 0.9$ eV. The key findings and conclusions did not show any qualitative change along with the variation of $U_{eff}$. **c** Formation energy of $V_O^{\cdot\cdot}|_{LuFe_2O_4}$ charged oxygen vacancies as a function of oxygen partial pressure and temperature.

substrates, as described in ref. 23, including e-beam induced Pt deposition. A 300 nm Pt, Cr or Ti protective layer was deposited on top of the film with e-beam evaporation using a Pfeiffer Vacuum Classic 500, at a rate of 1 Å/s. The characteristic needle-shaped specimens for APT were prepared with a Helios NanoLab DualBeam FIB as described by ref. 29. Cross-sectional transmission electron microscopy (TEM) specimens were prepared using an FEI Strata 400 FIB with a final milling step of 2 keV to reduce surface damage.

### Transmission electron microscopy
Selected specimens for APT were inspected to ensure adequate sample quality with TEM using a JEOL JEM-2100F Field Emission Electron Microscope operating at 200 kV. The high-resolution HAADF-STEM image in Fig. 1 was acquired on a 100-keV Nion UltraSTEM, a fifth-order aberration-corrected microscope. The lutetium distortions were quantified from HAADF-STEM images, as described in ref. 23

### Atom probe tomography
APT measurements were recorded with a Cameca LEAP 5000XS instrument, operating in laser-pulsed mode. Data were collected at cryogenic temperature ($T = 25$ K) with an applied bias between 2 kV and 10 kV. Laser pulses with 30 pJ energy and 250 kHz frequency were used, and the detection rate was set to 0.5%, i.e., 2 ions detected every 1000 pulse. The raw APT data was reconstructed into 3D datasets with the Cameca IVAS 3.6.12 software, using the voltage profile to determine the radial evolution. The image compression factor and field reduction factor were adjusted to make the thin film flat relative to the substrate. We note that the average oxygen concentration is lower than the nominal, which is common in APT, resulting from the detection loss of neutral O₂ molecules[27,30]. The charge-state-ratio (CSR) measured across the superlattice is presented in Fig. S3. The CSR shows no significant change in the electric field strength and, hence, the formation of O₂ neutrals is not expected to change throughout the superlattice.

### First-principles calculations of oxygen vacancy formation
To understand the tendency of formation of an oxygen vacancy ($V_O$) in the $(LuFeO_3)_9/(LuFe_2O_4)_1$ superlattice, we calculated the formation energy ($E_{V_O}^f$) of an oxygen vacancy as a function of temperature ($T$) and oxygen partial pressure ($p$) by considering the lowest energy crystal structure of the $(LuFeO_3)_3/(LuFe_2O_4)_1$ superlattice (obtained from DFT and STEM[23], see Fig. 3a). The formation of oxygen vacancies was studied by extracting one oxygen atom from the supercell of the superlattice. We used the following equation to calculate $E_{V_O}^f$ [31,32]:

$$E_{V_O}^f = E_{V_O} - E_0 + \mu_O + qE_F \tag{1}$$

where $E_{V_O}$ and $E_0$ represent the total energies of the supercell with and without an oxygen vacancy in a $q$ charge state, respectively. Here, the formation energy of an oxygen vacancy in a material depends on the oxygen chemical potential ($\mu_O$) and the electronic chemical potential ($E_F$). The chemical potential of oxygen atoms was calculated by the following equation[31,33]:

$$\mu_O(p,T) = \mu_O(p_0,T_0) + \mu_O(p_1,T) + \frac{1}{2}k_BT\ln\left(\frac{p}{p_1}\right) \tag{2}$$

Here, $\mu_O(p_0,T_0)$ represents the oxygen chemical potential at zero pressure ($p_0 = 0$) and zero temperature ($T_0 = 0$). According to the first-principles calculations, $\mu_O(p_0,T_0) = \frac{1}{2}E(O_2)$, where $E(O_2)$ denotes the total energy of an O₂ molecule. The second term, $\mu_O(p_1,T)$, which denotes the contribution of the temperature to the oxygen chemical potential at a particular pressure, was obtained from the experimental data[34]. The third term, $\frac{1}{2}k_BT\ln\left(\frac{p}{p_1}\right)$, represents the contribution of pressure to the chemical potential of oxygen. Here, $k_B$ is the Boltzmann constant. To calculate the formation energy of a charged (2+) vacancy, the total charge of the supercell was neutralized using jellium background. To incorporate the energy shift associated with

the jellium neutralization, we used the total energy of the positively charged perfect crystal (i.e., electrons were removed from the valence band maximum, and the jellium neutralization was included) as reference state. In the present study, we considered two kinds of oxygen vacancies, located in the Lu- ($V_O^{LuO_2}$) or Fe- ($V_O^{FeO}$) planes. We observed that the formation energy for a single oxygen defect in the Lu planes is generally higher than that in the Fe planes. Going beyond the superlattice structure in Fig. 3a (i.e., non-polar with asymmetric down-up-up and down-up-down Lu displacement patterns), we also calculated the formation energy of an oxygen vacancy for the polar $LuFe_2O_4$ and $LuFeO_3$ configurations, where Lu ions move symmetrically around both the Fe bi- and single-plane.

### Computational details

We calculated the total energies by performing first-principles calculations by employing the DFT method and the projector augmented plane-wave basis method as implemented in the VASP (Vienna Ab initio Simulation Package)[35,36]. The Perdew–Burke–Ernzerhof (PBE) form of the generalized gradient approximation (GGA) was used to calculate the exchange correlation functional[37]. A kinetic energy cut-off value of 500 eV and appropriate k-point meshes were selected so that total ground state energies were converged to $10^{-6}$ eV and the Hellman–Feynman forces were converged to 0.001 eV Å$^{-1}$. For each structure, coordinates of all atoms and lattice vectors were fully relaxed. The GGA + U method as developed by Dudarev et al.[38] was employed to deal with electron correlation in the Fe 3$d$ state. All calculations were performed by considering two values of $U_{eff} = U - J_H$, 4.5 and 6.5 eV, for the Fe 3$d$ states, where $U$ and $J_H$ represent the spherically averaged matrix elements of on-site Coulomb interactions[23,39,40]. We considered Lu 4$f$ states in the core. All the calculations of total energies were performed with antiferromagnetic and ferromagnetic collinear arrangement of Fe spins and without spin-orbit coupling. The stability range of $V_O^{..}|_{LuFe_2O_4}$ as a function of oxygen partial pressure and temperature were calculated, considering both oxygen-rich and oxygen-poor conditions (Fig. 3c). Concerning ferroelectricity, we note that the comparison of the optimized crystal structures before and after the formation of charged oxygen vacancies reveals a maximum reduction of the Lu displacement $\Delta d \sim 0.1$ Å. This displacement is not expected to have a measurable effect on the ferroelectric properties.

### Estimation of O vacancy density

Due to a change in unit cell composition at the $LuFe_2O_4$ layer, the O vacancy density cannot directly be extracted from the profile in Fig. 2. Instead, a simulation based on the ideal superlattice structure without any defect was made (solid line in Fig. 2). Using a DFT-based structure of the superlattice, the ideal atomic distribution was simulated. The atoms were then shifted around randomly to simulate the spatial resolution of the experiment, which was done with a Gaussian distribution with 0.6 nm standard deviation. This resolution value was found to best match the Fe and Lu distribution in the experimental data. A chemical profile across the simulated structure was then performed to get an expectation of an ideal superlattice structure. The difference between the simulated profile to the real data (shaded area in Fig. 2), fitted by a Gaussian profile, represents a measure for the oxygen vacancy concentration. A $V_O$ density was calculated by multiplying the $V_O$ concentration with the oxygen density from the simulated data. As the APT spatial resolution is not atomically sharp over the whole volume, the oxygen vacancy distribution is spread out over the measured width of the $LuFe_2O_4$ layer. Thus, for each measurement point, a $V_O$ density is measured, and after integrating across the interface a value for the total $V_O$ density is obtained. The change in $LuFe_2O_4$ composition is estimated by assuming all the oxygen vacancies are confined within a single unit cell. The 3D map of the oxygen depletion (Fig. 2c) is derived by displaying the chemical

composition in the vertical dimension within five $20 \times 20 \times 1.5$ nm$^3$ volumes. Chemical composition is converted into formula units (i.e., $LuFe_2O_{4+x}$) by measuring the local chemical composition in pixels of $2 \times 2$ nm$^2$. Note that the correction for the limited APT spatial resolution is not applied to these maps, and thus the measured chemical composition can appear lower than the calculated value from the 1D profile in Fig. 2b.

## Data availability

The data generated in this study have been deposited in an open-access repository and can be accessed at ref. 41. Further information, e.g., for plotting or analysis, can be provided upon request.

## Code availability

Computer codes used for simulations and data evaluation are available from the sources cited.

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

## Acknowledgements
We thank S. M. Selbach for very fruitful discussions and valuable input. The Research Council of Norway is acknowledged for the support to the Norwegian Micro- and Nano-Fabrication Facility, NorFab, project number 295864, the Norwegian Laboratory for Mineral and Materials Characterisation, MiMaC, project number 269842/F50, and the Norwegian Center for Transmission Electron Microscopy, NORTEM (197405/F50). K.A.H. and D.M. thank the Department of Materials Science and Engineering at NTNU for direct financial support. D.M.

acknowledges funding from the European Research Council (ERC) under the European Union's Horizon 2020 research and innovation program (Grant Agreement No. 863691) and further thanks NTNU for support through the Onsager Fellowship Program and NTNU Stjerne-programmet. Research at the Tokyo Institute of Technology is supported by the Ministry of Education, Culture, Sports, Science and Technology, Japan Grants-in-Aid No 24H00374 and JST-CREST (JPMJCR22O1). H.D. also acknowledges computational support from TSUBAME supercomputing facility. Film growth and electron microscopy characterization is supported by the US Department of Energy, Office of Basic Energy Sciences, Division of Materials Sciences and Engineering, under Award No. DE-SC0002334. The Electron Microscopy Facilities at the Cornell Center for Materials are supported through the NSF MRSEC program (DMR-1719875).

## Author contributions
K.A.H. conducted the APT experiments, including sample preparation and data analysis, supported by C.H., under the supervision of A.T.J.v.H and D.M. Samples by C.M.B., D.G.S., and J.A.M. and atomic-STEM by M.E.H. and D.A.M. DFT calculations were performed by H.D. D.M. initiated and coordinated the project. K.A.H. and D.M. wrote the manuscript. All authors discussed the results and contributed to the final version of the manuscript.

## Funding

## Competing interests
The authors declare no competing interests.
