## [Peer Review File · Nature Communications]

3D oxygen vacancy distribution and defect-property relations in an oxide heterostructureREVIEWER COMMENTS

Reviewer #1 (Remarks to the Author):

The authors describe the measurements of chemical elements using the atom probe tomography and electrical polarization using the electron microscopy. They correlate the polarization pattern with the so-called oxygen vacancy order. The conclusion is quite questionable, however. There are problems for both experimental and theoretical aspects. On the experimental aspect, there are seven LuFe₂O₄ layers, but only two layers are analyzed, making the presented results lacking statistical meaning. On the theoretical aspect, the charge state of oxygen vacancies is simply neglected, which is not acceptable for insulating oxides, in particular ferroelectric properties are major concerns in the manuscript.

Reviewer #2 (Remarks to the Author):

The oxygen vacancies have a very important impact on the properties of the oxide heterostructures, but quantifying their impact is a big challenge. In the present work, the atom probe tomography (ATP) combined DFT method was used to quantify the oxygen vacancies in the (LuFeO₃)₉/(LuFe₂O₄)₁ heterostructures. This work is exciting and suitable for the Journal after addressing the following comments:

In Fig. 1a, the arrow of polarization (P) is not consistent in the HAADF-STEM (head to head) and crystal structure (tail to tail), please confirm that.

In lines 161 and 163, there are two 'The latter', which is very confusing for the reader, please include more detailed information.

Line 163, "The latter carry a negative charge of about 12 μ C/cm², which is partially compensates the positive charge associated with the oxygen vacancies". Could the authors illuminate the origin of '12 μ C/cm²'?

Is there any experimental evidence of electron transfer from LuFe₂O_{4-x} layer to the middle LuFeO₃ layers as shown in Fig. 3d?

Reviewer #3 (Remarks to the Author):

This work is dedicated to further investigation of multiferroic nanocomposites LuFeO₃/LuFe₂O₄. In the current research the stoichiometry of nanocomposites was, for the first time, investigated at nanoscale with the technique of the Atomic Probe Tomography and this gives the great value to this research. The reported results discover the approach to easily (in comparison to the titrimetric methods) and quantitatively determine the concentration of oxygen vacancies. In fact, this seems to be the only technique applicable to thin films.

However, I have several concerns which I state below:

1) My first and greatest is that the concentration of the oxygen vacancies in LuFe₂O₄ is stated to correspond to the stoichiometry LuFe₂O_{3.5}, which means that all the iron atoms

have been reduced from +3 to +2. As far as I know, the greatest extent to which LuFe₂O₄ structure could be reversibly reduced is LuFe₂O_{3.935} [1]. If reduced further, it should decompose to Lu₂O₃ and FeO. Of course the reported data might be explained by the fact of epitaxial stabilization, however I would recommend a careful recalculation. Just in case.

2) The procedure of deriving the value of polarization of the LuFe₂O₄ layer from the vacancy concentration remains unclear to me. Despite the reported polarization value is quite close to the one that was reported previously [2], the commonly accepted conception is such that the charge order manifests in LuFe₂O₄ due to the valence ordering of iron ions in the stoichiometric LuFe₂O₄ (Fe⁺³/Fe⁺²=1) and not due to the presence of oxygen vacancies. In fact, it is thought that thanks to the variable valence of iron, LuFe₂O₄ remains neutral in chemical sense even when the vacancies or additional oxygen atoms are being inserted into the structure. In this scenario even if LuFe₂O_{3.5} does exist in the investigated material, charge ordering couldn't have existed in such compound. Thus it might be helpful to think about the systematic orientation of ferroelectric domains in h-LuFeO₃ layers from the interfacial strain point of view, since the ferroelectricity in this compound is known to be greatly affected by the nanoscale deformations.

3) Authors state that lutetium and iron APT-derived profiles perfectly correspond to those predicted for the ideal structure of LuFe₂O₄. However figures 2 and 2S clearly show that lutetium content is somewhat reduced and greatly shifted down from the sample surface compare to the ideal profile. I can't help but notice that the the minimum of the oxygen content is shifted in the opposite direction. I feel that it should not be regarded as a mere coincidence and must be discussed in the manuscript.

4) Like the authors I have no doubt in the fact that oxygen vacancies have great influence on the ferroelectric and magnetic properties of this material. However I feel that the information that is presented in this particular manuscript is not enough to lay down the assertion that the one-to-one correlation between the vacancy density and ferroelectric and magnetic properties. I would suggest authors to concentrate on the fact of application of the APT technique itself, not on the interrelation between ferroelectricity/magnetism and density of the oxygen vacancies since those still appear to be unclear.

In overall, I think that this research bears a great experimental importance demonstrating the application of the APT for the quantitative chemical composition profiling of thin-film nanocomposites. My personal recommendation is to publish it after a thorough revision.

[1] T. Sekine, T. Katsura, Phase equilibria in the system Fe-Fe₂O₃-Lu₂O₃ at 1200°C, J Solid State Chem. 17 (1976) 49–54. [https://doi.org/10.1016/0022-4596\(76\)90200-0](https://doi.org/10.1016/0022-4596(76)90200-0).

[2] N. Ikeda, H. Ohsumi, K. Ohwada, K. Ishii, T. Inami, K. Kakurai, Y. Murakami, K. Yoshii, S. Mori, Y. Horibe, H. Kitô, Ferroelectricity from iron valence ordering in the charge-frustrated system LuFe₂O₄, Nature. 436 (2005) 1136–1138. <https://doi.org/10.1038/nature04039>.

REVIEWER COMMENTS

We greatly appreciate the time and efforts the reviewers have invested in carefully reading and commenting on our work. In the following, we answer to all points raised by the three reviewers and, in accordance with their comments, have improved our article (for convenience, all changes are marked in the revised manuscript and SI in yellow).

Reviewer #1 (Remarks to the Author):

General remark: “The authors describe the measurements of chemical elements using the atom probe tomography and electrical polarization using the electron microscopy. They correlate the polarization pattern with the so-called oxygen vacancy order. The conclusion is quite questionable, however. There are problems for both experimental and theoretical aspects.”

General answer: We thank the reviewer for carefully reading the manuscript, reflecting on the details, and providing constructive feedback. We respond to the reviewer’s comments point by point in the following.

Remark 1.1: “On the experimental aspect, there are seven LuFe₂O₄ layers, but only two layers are analyzed, making the presented results lacking statistical meaning. “

Answer 1.1: We should start by saying that we took great care to present statistically meaning full results. For each of the analyzed layers, an area of about 20 x 20 nm² was analyzed, containing > 10⁵ atoms. This analysis gives statistically robust and meaningful results, clearly proving a significant enhancement in the density of oxygen defects in the LuFe₂O₄ layers relative to the LuFeO₃ layers (also reflected by the error bars in Figure 2a).

We specifically selected the two layers mentioned by the reviewer for the in-depth analysis as they are the most well-defined ones in the data set with a spacing of about 7 nm relative to the neighboring ones. This spatial separation allows for a reliable characterization of individual layers within the resolution of the APT experiment, without the risk of intermixing effects caused by close-by layers.

Importantly, the quantification of oxygen vacancies at the different interfaces – which is the main results of this work – is not to be confused with emergent layer-to-layer variations. The latter are not in the focus of this work, and we agree with the reviewer that their quantification would require the analysis of a much larger set of layers.

Motivated by comment reviewer’s comment, we measured additional samples, looking at other well-defined LuFe₂O₄ layers that allow for a reliable APT analysis (see Figure R1). Figure R1a,b shows the results obtained from two APT needles extracted at different positions from the same superlattice sample as discussed in the main text. We consistently observe a statistically significant accumulation of oxygen vacancies at the LuFe₂O₄ layers as reflected by the error bars. Furthermore, we expanded our investigations to LuFeO₃/LuFe₂O₄ superlattices with a larger separation between the LuFe₂O₄ layers. Here, the same effect is observed at the LuFe₂O₄ layers, i.e., an oxygen depletion and an increase in Lu and Fe concentration.

Thus, we can safely conclude that oxygen vacancies accumulate at the LuFe₂O₄ layers, and we emphasize that the extracted numbers are statistically meaningful and significant as indicated by the respective error bars.

Figure R1: Analyses of representative LuFe_2O_4 layers in two different superlattices: Data for $(\text{LuFeO}_3)_9/(\text{LuFe}_2\text{O}_4)_1$ is shown in **a** and **b**, and for $(\text{LuFeO}_3)_{15}/(\text{LuFe}_2\text{O}_4)_1$ in **c** and **d**. Dotted lines and symbols represent the experimental data, whereas solid lines represent concentrations expected for defect-free DFT-based models. All data sets consistently show oxygen depletion compared to the DFT-based models, corresponding to an accumulation of oxygen vacancies, accompanied by fluctuations in the Fe/Lu ratio.

Changes to the manuscript 1.1: In the revised manuscript, we clarify on page 6, line 124, that the results for each LuFe_2O_4 layer are statistically meaningful, writing “For each layer, more than 10^5 atoms are measured, over an area of $20 \times 20 \times 5 \text{ nm}^3$, providing local error estimates for the concentrations on the order of 0.3 at. %”. Furthermore, we expanded Supplementary Figure S2, where we present the new APT measurements as seen in Figure R1. We also explicitly mention now on page 6, line 144, that the quantification of sample-specific layer-to-layer variations go beyond the scope of this work. In Figure 2c, we increased the pixel volume over which we average to 6 nm^3 , containing more than 10^3 atoms each, giving statistically significant variations for lateral variations within the LuFe_2O_4 layers.

Remark 1.2: “On the theoretical aspect, the charge state of oxygen vacancies is simply neglected, which is not acceptable for insulating oxides, in particular ferroelectric properties are major concerns in the manuscript.”

Answer 1.2: We agree that this is an important aspect and that a careful analysis of the impact of the oxygen charge state improves the study. Hence, to identify differences for charged and neutral oxygen vacancies, we performed additional DFT calculations considering the lowest energy crystal structure of the $(\text{LuFeO}_3)_3/(\text{LuFe}_2\text{O}_4)_1$ superlattice (obtained from DFT and STEM¹, see Figure R2a). The defect formation energy for a single oxygen vacancy in a q charge state is defined as $\Delta E_f = E_{V_O} - E_0 + \mu_O + qE_F$, where μ_O and E_F denote the chemical potential of oxygen and the electronic chemical potential (Fermi energy), respectively (computational details are provided in Methods). The calculated total energies of the supercell with and without an oxygen vacancy are denoted as E_{V_O} and E_0 , respectively. The additional DFT results are presented in Figure R2b,c, showing that the formation energy for a single oxygen defect in the Lu planes is generally higher than in Fe planes. Furthermore, the formation of charged defects is energetically favorable in LuFe_2O_4 compared to LuFeO_3 , yielding an energy reduction of 64 meV. Compared to neutral defects in LuFe_2O_4 , $V_{\text{O}}^X|_{\text{LuFe}_2\text{O}_4}$, the formation of charged

defects, $V_{\text{O}}^{\cdot\cdot}|_{\text{LuFe}_2\text{O}_4}$, is about 1 eV lower in energy. The stability range of $V_{\text{O}}^{\cdot\cdot}|_{\text{LuFe}_2\text{O}_4}$ as a function of oxygen partial pressure and temperature is displayed in Figure R2c, considering both oxygen-rich and oxygen-poor conditions.

Going beyond the superlattice structure in Figure R2a (i.e., non-polar with asymmetric down-up-up and down-up-down Lu displacement patterns), we also calculated formation energy of an oxygen vacancy at the polar LuFe_2O_4 and LuFeO_3 configurations, where Lu ions move symmetrically around both the Fe bi- and single- plane. The calculations reveal:

- (i) Irrespective of the LuFe_2O_4 layer being polar or non-polar, the formation of $V_{\text{O}}^{\cdot\cdot}|_{\text{LuFe}_2\text{O}_4}$ is always the energetically most favorable defect.
- (ii) The stability of a charged oxygen vacancy is significantly enhanced compared to a neutral oxygen vacancy in the non-polar LuFe_2O_4 layer compared to the polar counterpart (0.02 \rightarrow 1.00 eV).

Thus, in agreement with the APT experiments, the DFT results show that an accumulation of oxygen vacancies in the LuFe_2O_4 layer is energetically favorable.

Concerning ferroelectricity, we note that the comparison of the optimized crystal structures before and after the formation of charged oxygen vacancies reveals a maximum reduction of the Lu displacement $\Delta d \sim 0.1 \text{ \AA}$. This displacement is not expected to have a measurable effect on the ferroelectric properties.

Figure R2: Defect formation energy for oxygen vacancies in LuFe_2O_4 and LuFeO_3 . **a**, Crystal structure of the $(\text{LuFeO}_3)_3/(\text{LuFe}_2\text{O}_4)_1$ superlattice, demonstrating tail-to-tail and head-to-head ferroelectric domain walls. **b**, The formation energy of oxygen vacancy at the oxygen-rich limit and for $E_F = 0$ (which corresponds to valence band maximum). $V_{\text{O}}^{\cdot\cdot}|_{\text{LuFe}_2\text{O}_4}$ ($V_{\text{O}}^{\cdot\cdot}|_{\text{LuFeO}_3}$) and $V_{\text{O}}^{\times}|_{\text{LuFe}_2\text{O}_4}$ ($V_{\text{O}}^{\times}|_{\text{LuFeO}_3}$) corresponding to oxygen vacancy in 2+ and neutral charge state at the LuFe_2O_4 (LuFeO_3) layer, respectively. We conducted calculations for $U_{\text{eff}} = 6.5 \text{ eV}$ at the Fe 3d orbital and the corresponding band gap is $E_g = 0.9 \text{ eV}$. **c**, The formation energy of $V_{\text{O}}^{\cdot\cdot}|_{\text{LuFe}_2\text{O}_4}$ charged oxygen vacancy as a function of oxygen partial pressure and temperature.

In summary, the extended DFT analysis clarifies that an oxygen vacancy formation in the LuFe_2O_4 layers is energetically favorable compared to the LuFeO_3 layers. The energy costs for charged oxygen vacancies are lower than for neutral ones, and the defect formation has no significant impact on the polar displacements that give rise to ferroelectricity.

Changes to the manuscript 1.2: The additional DFT results are included in the revised main manuscript, where Figure R2 replaces the previous Figure 3. We specifically address the oxygen charge state on line 150 on page 7.

Reviewer #2 (Remarks to the Author):

General remark: “The oxygen vacancies have a very important impact on the properties of the oxide heterostructures, but quantifying their impact is a big challenge. In the present work, the atom probe tomography (APT) combined DFT method was used to quantify the oxygen vacancies in the $(\text{LuFeO}_3)_9/(\text{LuFe}_2\text{O}_4)_1$ heterostructures. This work is exciting and suitable for the Journal after addressing the following comments”

General answer: We greatly appreciate the reviewer’s very positive feedback and address all her/his constructive technical remarks in our point-by-point below.

Remark 2.1: In Fig.1a, the arrow of polarization (P) is not consistent in the HAADF-STEM (head to head) and crystal structure (tail to tail), please confirm that.

Answer 2.1: We thank the reviewer for pointing this out. The direction was indeed wrong and has been corrected in the revised manuscript.

Remark 2.2: In lines 161 and 163, there are two ‘The latter’, which is very confusing for the reader, please include more detailed information.

Answer 2.2: Agreed. We have revised the sentence accordingly.

Remark 2.3: Line 163, “The latter carry a negative charge of about $12\mu\text{C}/\text{cm}^2$, which is partially compensates the positive charge associated with the oxygen vacancies”. Could the authors illuminate the origin of ‘ $12\mu\text{C}/\text{cm}^2$ ’?

Answer 2.3: The 180° polarization reorientation at the interface yields a bound charge, which is $2P = 13\mu\text{C}/\text{cm}^2$ ($P = 6.5\mu\text{C}/\text{cm}^2$, ref. ²). We thank the reviewer for bringing the typo to our attention.

Changes to manuscript 2.3: We have corrected the sentence and added a reference for the spontaneous polarization.

Remark 2.4: Is there any experimental evidence of electron transfer from $\text{LuFe}_2\text{O}_{4-x}$ layer to the middle LuFeO_3 layers as shown in Fig. 3d?

Answer 2.4: Our APT experiments cannot show such electron transfer directly. Going beyond previous EELS experiments, however, by applying APT we are able to measure an oxygen vacancy accumulation at the LuFe_2O_4 layers. This accumulation – together with the reported stability of head-to-head walls in the LuFeO_3 layers – represents first experimental evidence for the electron-transfer hypothesis proposed in ref. ¹, and it is fully consistent with published first-principle calculations¹ and our new DFT results, revealing that charged oxygen vacancies are energetically favorable in the LuFe_2O_4 layer (please see answer 1.2 for details).

Changes to manuscript 2.4: In the revised manuscript, we clarify on page 8, line 170, that the electron transfer has not been measured directly and discuss how it is inferred from the accumulation of oxygen vacancies and DFT calculations, the stability of head-to-head walls in the LuFeO_3 layers, and the system’s propensity to form charged oxygen defects in the LuFe_2O_4 layers.

Reviewer #3 (Remarks to the Author):

General remark: “This work is dedicated to further investigation of multiferroic nanocomposites LuFeO₃/LuFe₂O₄. In the current research the stoichiometry of nanocomposites was, for the first time, investigated at nanoscale with the technique of the Atomic Probe Tomography and this gives the great value to this research. The reported results discover the approach to easily (in comparison to the titrimetric methods) and quantitatively determine the the concentration of oxygen vacancies. In fact, this seems to be the only technique applicable to thin films.

In overall, I think that this research bears a great experimental importance demonstrating the application of the APT for the quantitative chemical composition profiling of thin-film nanocomposites. My personal recommendation is to publish it after a thorough revision.”

General answer: We greatly appreciate the reviewer’s positive feedback and that she/he feels that the APT experiments give great value to the research on nanocomposites. In the following, we respond to all the reviewer’s comments.

Remark 3.1: My first and greatest is that the concentration of the oxygen vacancies in LuFe₂O₄ is stated to correspond to the stoichiometry LuFe₂O_{3.5}, which means that all the iron atoms have been reduced from +3 to +2. As far as I know, the greatest extent to which LuFe₂O₄ structure could be reversibly reduced is LuFe₂O_{3.935} [1]. If reduced further, it should decompose to Lu₂O₃ and FeO. Of course the reported data might be explained by the fact of epitaxial stabilization, however I would recommend a careful recalculation. Just in case.

Answer 3.1: We agree that this is an important point and we considered it carefully. Please note that the work by Semine *et al.* addresses the equilibrium thermodynamics of a homogenous material, implying an ergodic state where the global ground state can be reached. In that case, no spatial gradients exist in the chemical potential of any species and, opposite to our material, the oxygen vacancy concentration is uniform.

This situation is very different from the (LuFeO₃)₃/(LuFe₂O₄)₁ superlattice we study, which is not in a global equilibrium state and was not grown by an equilibrium method. For example, LuFeO₃ is not a stable bulk compound and cannot be synthesized bulk material. In addition to the superlattice’s non-equilibrium state, epitaxy – as mentioned by the reviewer – potentially plays an additional role in stabilizing LuFe₂O_{3.5} as the mechanical clamping imposed by the substrate in epitaxy suppresses nucleation of a secondary phase. We are thus confident, that the superlattice can locally accommodate a reduction to LuFe₂O_{3.5} without decomposing into Lu₂O₃ and FeO.

We also double-checked that our analysis is correct: The reported stoichiometry LuFe₂O_{3.935} for bulk corresponds to a decrease in oxygen concentration of 0.4 at.% relative to LuFe₂O₄³. In contrast, we consistently measure a substantially larger drop for the LuFe₂O₄ layers in both (LuFeO₃)₉/(LuFe₂O₄)₁ and (LuFeO₃)₁₅/(LuFe₂O₄)₁ superlattices, approaching a peak value of about 1 at.% (see, e.g., Figure 2b). This difference already demonstrates that, locally, the reduction exceeds the literature value reported for bulk systems.

Assuming that this drop is associated with a one-unit-cell thick LuFe₂O₄ layer (which is justified based on the excellent agreement between the model and the measured Lu and Fe concentration profiles), we find LuFe₂O_{3.835}. This estimate, however, does not account for the resolution of the APT experiment, which leads to a broadening of the minimum in the oxygen concentration profile (to around 2 nm), whereas TEM shows that the actual thickness of the LuFe₂O₄ layer is about 0.5 nm. To correct for this effect, we integrate over the oxygen concentration profile, which gives the stoichiometry LuFe₂O_{3.5}.

Changes to the manuscript 3.1: For transparency, we now give more details in the methods to clarify how the local stoichiometry was determined, in addition to a mention in the main text (page 6, line 133). Furthermore, we explicitly discuss that the values locally measured for the unit-cell-thick LuFe₂O₄ layers exceed the limit previously reported for bulk systems, which is consistent with the

inherent metastable state of a superlattice in local - but not global - equilibrium and, possibly, further promoted by epitaxial stabilization.

Remark 3.2.1: The procedure of deriving the value of polarization of the LuFe₂O₄ layer from the vacancy concentration remains unclear to me.

Answer 3.2.1: The ferroelectric polarization we are referring to is the spontaneous polarization of the (LuFeO₃)₃ layers ($P = 6.5 \mu\text{C}/\text{cm}^2$, ref. ²). It is established that the polarization vectors form a tail-to-tail configuration ($\leftarrow\rightarrow$) at the (LuFe₂O₄)₁ layers, leading to a polar discontinuity, equivalent to a negative bound $2P = 13 \mu\text{C}/\text{cm}^2$, that requires screening. This is consistent with the measured accumulation of oxygen vacancies (= positive charges).

Changes to manuscript 3.1: We have now made it more clear in the discussion on page 8, line 174, that the interface bound charges at the (LuFe₂O₄)₁ layers originate from the (LuFeO₃) layers.

Remark 3.2.2: Despite the reported polarization value is quite close to the one that was reported previously [2], the commonly accepted conception is such that the charge order manifests in LuFe₂O₄ due to the valence ordering of iron ions in the stoichiometric LuFe₂O₄ (Fe⁺³/Fe⁺²=1) and not due to the presence of oxygen vacancies. In fact, it is thought that thanks to the variable valence of iron, LuFe₂O₄ remains neutral in chemical sense even when the vacancies or additional oxygen atoms are being inserted into the structure. In this scenario even if LuFe₂O_{3.5} does exist in the investigated material, charge ordering couldn't have existed in such compound.

Answer 3.2.2: We feel that there is misunderstanding concerning the origin of ferroelectricity in the (LuFeO₃)₁₅/(LuFe₂O₄)₁ superlattice. Please note that we do not claim that LuFe₂O₄ is ferroelectric. As reported in Mundy *et al.*¹, ferroelectricity arises from the geometrically driven improper ferroelectric order in LuFeO₃, manifesting in the characteristic up-up-down = + P and down-down-up = - P displacement patterns of the Lu atoms we show in Figure 1a. The one unit-cell-thick LuFe₂O₄ layers represent the magnetic constituent in the artificial multiferroic superlattice.

Remark 3.2.3: Thus it might be helpful to think about the systematic orientation of ferroelectric domains in h-LuFeO₃ layers from the interfacial strain point of view, since the ferroelectricity in this compound is known to be greatly affected by the nanoscale deformations.

Answer 3.2.3: It is correct that strain gradients can, in principle, lead to a movement of the topological vortices in hexagonal LuFeO₃ and affect the domain structure (see, e.g., Holtz *et al.*⁴, where related effects for isostructural hexagonal manganites are discussed). While we cannot exclude that strain plays a role for the domain formation, it cannot explain how the fully charged head-to-head and tail-to-tail 180° walls are stabilized. The associated diverging electrostatic potentials are energetically extremely costly and require screening, which we demonstrated is achieved via the accumulation of oxygen vacancies, clarifying how the system stabilizes its unusual domain structure.

Remark 3.3: Authors state that lutetium and iron APT-derived profiles perfectly correspond to those predicted for the ideal structure of LuFe₂O₄. However figures 2 and 2S clearly show that lutetium content is somewhat reduced and greatly shifted down from the sample surface compare to the ideal profile. I can't help but notice that the minimum of the oxygen content is shifted in the opposite direction. I feel that it should not be regarded as a mere coincidence and must be discussed in the manuscript.

Answer 3.3: The reviewer is correct that there are shifts in the O and Lu concentration profiles, which we attribute to delayed or preferential evaporation effects of ionic species^{5,6} during the APT analysis. It is well-established that elements which require a higher electric field to be evaporated (high-field elements, here: Lu) are retained on the surface, whereas elements that require a lower electric field (low-field elements, here: O) evaporate earlier. This effect leads to biases in the reconstruction, and it is visible as small spatial shifts of a few Ångström in the compositional profile.

To corroborate our conclusion and gain additional insight, we analyze the field evaporation behavior of the superlattice by considering the charge-state-ratio (CSR) evolution as shown in Figure R3. The CSR is directly related to the electric field strength during the APT analysis and, hence, sensitive to field evaporation artefacts.

Figure R3 shows an increased ionic density at the LuFe_2O_4 layers, which is well-known to arise in APT due to an overall lower field evaporation (we note that this increase does not directly translate into a higher ionic density). Most importantly, the LuFe_2O_4 layers have no detectable impact on the CSR profile, which reflects that the electric field strength is largely constant throughout the superlattice. The data shows that there is no substantial difference in the field evaporation of the LuFeO_3 and LuFe_2O_4 layers, excluding field evaporation artefacts beyond subtle atomic shifts, consistent with the systematic shifting of low-field O and high-field Lu seen in Figure 2 and S2, as well as the new datasets in Figure R1.

Figure R3: CSR analysis of the superlattice system, using the Fe charge state (black line). The vertical dashed lines indicate the position of the LuFe_2O_4 layers, obtained by considering the ionic density profile (red line), which can detect very subtle changes to the electric field evaporation criteria. Importantly, the CSR analysis does not identify any substantial changes to the electric field strength, which rules out any major artefacts stemming from changes to the field evaporation criteria.

Changes to the manuscript 3.3: In the revised manuscript, we comment on the field-evaporation-related shift on page 6, line 131, providing an extended explanation along with the new CSR data in Supplementary Figure S3.

Remark 3.4: Like the authors I have no doubt in the fact that oxygen vacancies have great influence on the ferroelectric and magnetic properties of this material. However I feel that the information that is presented in this particular manuscript is not enough to lay down the assertion that the one-to-one correlation between the vacancy density and ferroelectric and magnetic properties. I would suggest authors to concentrate on the fact of application of the APT technique itself, not on the interrelation between ferroelectricity/magnetism and density of the oxygen vacancies since those still appear to be unclear.

[1] T. Sekine, T. Katsura, Phase equilibria in the system Fe-Fe₂O₃-Lu₂O₃ at 1200°C, J Solid State Chem. 17 (1976) 49–54. [https://doi.org/10.1016/0022-4596\(76\)90200-0](https://doi.org/10.1016/0022-4596(76)90200-0).

[2] N. Ikeda, H. Ohsumi, K. Ohwada, K. Ishii, T. Inami, K. Kakurai, Y. Murakami, K. Yoshii, S. Mori, Y. Horibe, H. Kitô, Ferroelectricity from iron valence ordering in the charge-frustrated system LuFe₂O₄, Nature. 436 (2005) 1136–1138. <https://doi.org/10.1038/nature04039>.

Answer 3.4: Following the reviewer’s suggestion, we have removed the statements about a “one-to-one correlation” between oxygen defects and the electric and magnetic order. However, we respectfully disagree that the “*on the interrelation between ferroelectricity/magnetism and density of the oxygen vacancies since those still appear to be unclear*”. Our APT measurements unambiguously show an accumulation of oxygen vacancies at the one-unit-cell thick LuFe_2O_4 layers, providing first experimental evidence for how the system stabilizes the unusual ferroelectric order in the LuFeO_3 layers at the level of domains. Furthermore, it is clear that the associated changes in the Fe oxidation state impacts the magnetic properties of the LuFe_2O_4 as discussed in ref. ⁷. Thus, we feel that it is safe to conclude that the measured oxygen vacancy density is connected with the multiferroic properties of $(\text{LuFeO}_3)_9/(\text{LuFe}_2\text{O}_4)_1$.

Changes to the manuscript 3.4: We have revised the text accordingly, now putting more weight on the APT measurements themselves. Parts of the discussion on the correlation between ferroic ordering and defects have been moved to the outlook section.

References

1. Mundy, J. A. *et al.* Atomically engineered ferroic layers yield a room-temperature magnetoelectric multiferroic. *Nature* **537**, 523–527 (2016).
2. Jeong, Y. K., Lee, J. H., Ahn, S. J. & Jang, H. M. Epitaxially constrained hexagonal ferroelectricity and canted triangular spin order in LuFeO_3 thin films. *Chem. Mater.* **24**, 2426–2428 (2012).
3. Sekine, T. & Katsura, T. Phase equilibria in the system $\text{Fe Fe}_2\text{O}_3 \text{ Lu}_2\text{O}_3$ at 1200°C. *J. Solid State Chem.* **17**, 49–54 (1976).
4. Holtz, M. E. *et al.* Dimensionality-induced change in topological order in multiferroic oxide superlattices. *Phys. Rev. Lett.* **126**, (2021).
5. Hunnestad, K. A. *et al.* Correlating laser energy with compositional and atomic-level information of oxides in atom probe tomography. *Mater. Charact.* **203**, 113085 (2023).
6. Vurpillot, F., Bostel, A., Cadel, E. & Blavette, D. The spatial resolution of 3D atom probe in the investigation of single-phase materials. *Ultramicroscopy* **84**, 213–224 (2000).
7. Fan, S. *et al.* Site-specific spectroscopic measurement of spin and charge in $(\text{LuFeO}_3)_m/(\text{LuFe}_2\text{O}_4)_1$ multiferroic superlattices. *Nat. Commun.* **11**, 1–9 (2020).

REVIEWERS' COMMENTS

Reviewer #1 (Remarks to the Author):

In the revised version, the authors have added experimental and computational data that nicely address the concerns raised by this reviewer about the first version.

There is a term that needs to be revised, including that in the title. The authors used “oxygen vacancy order”, but there is only “accumulation” of oxygen vacancies, not ordering.

Reviewer #2 (Remarks to the Author):

The authors have addressed my comments.

Reviewer #3 (Remarks to the Author):

I heartily thank the authors for exhaustive answers to all the questions I had laid down. The authors demonstrate a comprehensive knowledge of the APT technique and provide convincing arguments defending their point. The provided results give another spectacular evidence of how defects in solids may give rise to interesting charge meta structures. What is more important in my opinion - this research provides a demonstration of effective investigation route of such structures with the combination of HRTEM and APT.

It is still hard for me to accept the fact of the $\text{LuFe}_2\text{O}_{3.5}$ average stoichiometry, but the reasoning provided by the authors seems congruent and sound. Therefore yet another breakthrough of this research is the demonstration of stabilization of LuFe_2O_4 phase in the extreme state.

Therefore I conclude that this article is worthy of being published in the Nature Communications journal.